# Current applications and outcomes of AI-driven adaptive learning systems in physical rehabilitation science education: A scoping review protocol

Oyindolapo O. Komolafe[1]*, Jannatul Mustofa[1], Mark J. Daley[2], David Walton[1], Andrews Tawiah[1]

1 School of Physical Therapy, Faculty of Health Sciences, Western University, London, Ontario, Canada,
2 Computer Science, Western University, London, Ontario, Canada

* okomola@uwo.ca

**Data availability statement:** No datasets were generated or analysed during the current study.

## Abstract

**Rationale** Integrating artificial intelligence (AI) into education has introduced transformative possibilities, particularly through adaptive learning systems. Rehabilitation science education stands to benefit significantly from the integration of AI-driven adaptive learning systems. However, the application of these technologies remains underexplored. Understanding the current applications and outcomes of AI-driven adaptive learning in broader healthcare education can provide valuable insights into how these approaches can be effectively adapted to enhance multimodal case-based learning in Rehabilitation Science education.

**Methods** The scoping review is based on the Joanne Briggs Institute (JBI) framework. It is reported according to the Preferred Reporting Items for Systematic Reviews and Meta-Analyses extension for Scoping Reviews (PRIMSA-ScR). A comprehensive search strategy will be used to find relevant papers in Scopus, PubMed, CINAHL, Education Resources Information Center (ERIC), Association for Computing Machinery (ACM), ProQuest Education Journal, Web of Science, ProQuest Dissertations & Theses Global, and IEEE Digital Library. This review will include all types of studies that describe or evaluate our outcomes of interest: AI models used, learning and teaching methods, effective implementation, outcomes, and challenges of ALS's in rehabilitation health science education. Data will be extracted using a pre-piloted data extraction sheet and synthesized narratively to identify themes and patterns.

**Discussion** This scoping review will synthesize the applications of AI models in rehabilitation science education. It will provide evidence for educators, healthcare professionals, and policymakers to incorporate AI into educational curricula effectively. The protocol is registered on Open Science Framework registries at https://osf.io/e46s3.

All relevant data from this study will be made available upon study completion.

**Funding:** The author(s) received no specific funding for this work.

**Competing interests:** The authors have declared that no competing interests exist.

## Introduction

As Artificial Intelligence (AI) transforms various sectors, its application in education has garnered significant attention [1]. Adaptive Learning is an educational approach that uses technology to provide personalized learning experiences tailored to individual needs, specific learning patterns, knowledge levels, and progress [2,3]. Adaptive learning systems use several AI techniques such as machine learning (ML), natural language processing (NLP) [4], reinforcement learning, and predictive analytics [5] to deliver content that is most relevant to the learner at any given point in their educational journey. Adaptive learning technologies—which use AI to modify the learning process based on real-time analysis of student performance [6] — have been applied in numerous educational settings, including K-12 education, higher education, and professional development programs [7]. Adaptive learning systems (ALS's) can diagnose knowledge gaps, predict learning trajectories, and offer personalized instruction to optimize learning outcomes. In doing so, they address the limitations of traditional, one-size-fits-all instructional methods [8].

In this context, education is defined as a holistic process that includes not only the delivery of content but also learner engagement, ongoing evaluation, formative and summative assessment, and feedback [9–13]. This comprehensive understanding is essential when examining adaptive learning systems, as their personalization capabilities are driven by continuous analysis of student performance data to inform real-time instructional adjustments.

Physical Rehabilitation science is a multidisciplinary field that uses conservative, non-surgical interventions to help people regain or maintain their ability to move and navigate their environment. This includes a wide range of techniques to improve strength, flexibility, mobility, and coordination, all to enhance functional independence [14–16]. The disciplines considered in Physical Rehabilitation for this context are: physical therapy, occupational therapy, orthotics, prosthetics, chiropractic, recreation therapy, sports therapy, rehabilitation counseling, kinesiology, audiology, speech-language pathology, and vocational therapy. Physical rehabilitation science education involves a unique blend of theoretical and practical learning. In rehabilitation science, students must develop both cognitive understanding and clinical decision-making skills, often requiring personalized approaches to bridge theoretical knowledge with real-world clinical applications [17,18]. Translating theoretical knowledge into clinical skills requires adaptive learning experiences that can respond to individual student needs, help develop critical thinking, and improve clinical decision-making. Traditionally, this education has been delivered through lectures, hands-on clinical training, and case-based learning, however, the increasing demand for accessible, flexible, and scalable education models, coupled with advancements in AI, has opened the door to innovative pedagogical approaches. One such innovation is the development of AI-driven adaptive learning systems, which have the potential to revolutionize how physical rehabilitation science students acquire knowledge and skills.

In this context, ALS's has the potential to revolutionize rehabilitation science education by providing personalized case-based learning experiences that mirror real-life clinical scenarios. These systems can allow learners to engage with virtual patients, simulate treatment plans, and receive immediate feedback on their decisions, thereby promoting a deeper understanding of physical rehabilitation principles and techniques. Moreover, ALS's can address the diverse learning needs of students by adapting the complexity of case studies, providing tailored resources, and allowing learners to progress at their own pace. AI-driven adaptive learning systems are designed to personalize the educational experience by tailoring content [1], feedback, and assessments based on individual student needs, progress, and learning styles.

Despite the potential of AI-driven adaptive learning and the growing body of evidence supporting the effectiveness of AI-driven adaptive learning systems in education, there is limited understanding of their current applications and outcomes within physical rehabilitation science education specifically [19] as current literature often focuses on using these systems in general education or healthcare more broadly. This scoping review seeks to map the landscape of AI-driven adaptive learning systems in this field and identify gaps, challenges, and opportunities for future research and practice. By mapping the current landscape, this review will provide valuable insights into the potential for AI-driven adaptive learning systems to enhance rehabilitation science education and inform future research and pedagogical practices.

The findings from this review will also benefit educators and curriculum developers in physical rehabilitation science, helping them make informed decisions about integrating AI-driven adaptive learning technologies into their programs. Additionally, by identifying challenges and opportunities, this review aims to contribute to the broader discourse on the role of AI in healthcare education, ultimately leading to better-prepared physical rehabilitation professionals and improved patient outcomes.

## Objective and research questions

The primary aim of this proposed scoping review is to identify, map, and synthesize the current applications of AI models in physical rehabilitation science education. Additionally, it seeks to identify gaps in the existing literature and provide recommendations for areas of future research and educational practice improvements within physical rehabilitation science education.

Specifically, the review will focus on identifying how AI-driven adaptive learning systems are being implemented in physical rehabilitation science education, examining the reported educational outcomes such as student engagement, knowledge acquisition, skill development, clinical reasoning, and overall performance, and highlighting the key challenges in implementing these systems while identifying areas for further research.

Research questions of the scoping review are:

1. What types of AI models have been utilized, and how are they being applied to enhance teaching and learning in physical rehabilitation science education?
2. What outcomes, challenges, and limitations are associated with using AI models in physical rehabilitation science education?
3. What gaps exist in the current literature, and what future directions are recommended to optimize AI integration in rehabilitation science education?

## Method

The scoping review will follow guidance from the Joanna Briggs Institute (JBI) Manual for Evidence Synthesis on Scoping Reviews [20] since the goal is to identify, map, and synthesize the current applications of AI models in rehabilitation science education. Using the JBI framework ensures a rigorous and systematic approach. To establish the quality of our protocol, the protocol is reported according to the Preferred Reporting Items for Systematic Reviews and Meta-Analyses extension for Scoping Reviews (PRISMA-SCR)[21]. This protocol is registered on OSF: https://osf.io/e46s3

### Eligibility criteria

The PCC (Population, Concept, and Context) framework informs the eligibility criteria.

**Inclusion criteria. Population**: The review will include studies involving students, educators, or professionals in the education of physical rehabilitation science, such as physical therapy, occupational therapy, orthotics, prosthetics, chiropractic, recreation therapy, sports therapy, rehabilitation counseling, kinesiology, audiology, speech-language pathology, and vocational therapy. Additionally, studies focused on educating healthcare professionals or students in physical rehabilitation-related disciplines will be considered.

**Concept**: Eligible studies must explore the application of AI models in enhancing teaching and learning within rehabilitation science education. They should report on outcomes of AI use, such as learning effectiveness, student engagement, or skill development, and identify challenges, limitations, or gaps in integrating AI into educational practices for rehabilitation science.

**Context**: The review will include studies conducted in educational settings for rehabilitation science, encompassing formal academic settings (e.g., universities, colleges) and professional training programs, both pre-professional and post-professional. Studies focusing on face-to-face, online, or blended learning environments will be included. Publications must be peer-reviewed or part of conference proceedings or grey literature, with a focus on AI in education, and published between January 2019 and December 2024.

**Language**: Only publications written in English will be included.

**Exclusion criteria. Population**: Studies focused on patients undergoing rehabilitation or using AI exclusively in rehabilitation treatment settings without an educational component will be excluded. Additionally, studies that do not involve students, educators, or professionals in rehabilitation science education will not be considered.

**Context**: Studies that do not specifically address the application of AI models in education or that focus solely on traditional, non-AI educational methods will be excluded. Similarly, studies that center solely on administrative AI applications without addressing any educational outcomes will not be included.

**Language**: Publications not written in English will be excluded.

## Types of sources

We will search the following sources: Scopus, PubMed, CINAHL, Education Resources Information Center (ERIC), Association for Computing Machinery (ACM), ProQuest Education Journal, Web of Science, ProQuest Dissertations & Theses Global, and IEEE Digital Library. We will search for studies from January 2019 and December 2024. Manually searching the reference list of identified studies will identify any further articles that meet eligibility criteria. Furthermore, we will review grey literature (registered protocols, conferences, and studies under process) and consult with artificial intelligence experts to identify potential studies.

## Search strategy and information sources

The search strategy will be developed with a research librarian (DL). The draft search strategy is presented in S1 Appendix. The search strategy combines structure database-specific subject headings (as available) and keywords/ synonyms of the following concepts:

- AI models
- Adaptive learning systems
- Physical Rehabilitation Science
- Education

Search terms within a concept will be connected with the boolean operator 'OR' while separate concepts will be connected with 'AND', while search terms within each concept will be combined using 'OR'. The search terms will be tailored to each database. To minimize publication bias, grey literature sources (conference proceedings and theses) will also be searched to identify studies of relevance to this review. Similarly, to avoid missing any relevant literature, we will also search the reference lists of included studies and those of relevant systematic reviews.

**Data extraction.** The data shown in Table 1 will be extracted from the literature.

## Data extraction process

A tailored data extraction form will be developed to systematically capture information from each study relevant to the research questions of this scoping review. The form will include variables such as study characteristics (e.g., authors, year of publication, country, and study

**Table 1. Data extraction Tool**

| Variable | Description |
|---|---|
| Study identification | Authors' names |
| | Year of publication |
| | Title of the study |
| | Journal name |
| Study characteristics | Type of Study (e.g., empirical research, review, case study) |
| | Study design (e.g., randomized control trial, cohort study, qualitative study) |
| | Sample size |
| | Setting (e.g., educational institution, clinical environment) |
| | Country where study took place |
| | Rehabilitation and Physical health program type |
| | Educational level of participants |
| AI Model Used | Description of the AI model(s) implemented (e.g., machine learning algorithms, adaptive learning technologies) |
| | Specific features of the AI system utilized (e.g., personalized feedback, real-time analytics) |
| Systems | Supporting systems used with the AI models |
| | Deployment Platforms |
| | Any underlying theories used in the study |
| Application in Education | How the AI Model was integrated into Rehabilitation Science Education (e.g., course, curriculum, specific training program) |
| | Teaching methods employed (e.g., blended learning, simulation-based learning) |
| Reported Outcomes | Student engagement (measured through surveys, assessments, etc.) |
| | Knowledge acquisition (test scores, performance metrics) |
| | Skill development (clinical skills assessment, practical evaluations) |
| | Clinical reasoning (decision-making assessments, case studies) |
| | Overall performance (final grades, course completion rates) |
| Implementation Challenges | Key challenges encountered in implementing the AI-driven systems (e.g., technical issues, resistance from faculty/students, resource limitations) |
| | Strategies used to overcome these challenges |
| | Ethical challenges |
| Research Gaps | Identified gaps in the existing literature related to AI applications in rehabilitation education |
| | Areas needing further investigation or development |
| Future Research Directions | Conclusions |
| | Recommendations provided by the authors for future research |
| | Suggested enhancements to educational practices involving AI |
| Funding and Conflicts of Interest | Source of funding for the study |
| | Any potential conflicts of interest disclosed by the authors |

design), the type of AI models used, the context of their application in rehabilitation science education, reported educational outcomes, challenges encountered, gaps in the literature, and recommendations for future research. The form will be designed to accommodate both quantitative and qualitative data, depending on the type of studies included. Before full-scale data extraction begins, the form will be piloted by the research team using a small sample of included studies. This pilot phase will help ensure that all relevant data are captured effectively and consistently across different types of studies. During this phase, any ambiguities in the form will be identified, and modifications will be made to improve clarity and usability. The pilot testing will also help ensure that the reviewers have a common understanding of the variables to be extracted. Once the data extraction form has been finalized, two reviewers will independently extract data from each included study. This independent process aims to minimize bias and ensure that no important details are overlooked. The independent extraction will also facilitate cross-checking and validation of the extracted data. After the independent extraction, the reviewers will compare their extracted data. Any discrepancies in the data will be discussed and resolved through consensus. In cases where agreement cannot be reached, a third reviewer will be consulted to adjudicate and make the final decision. This process will ensure that the extracted data is accurate and reflects the content of the original studies. The extracted data will be organized in a tabular format. This will facilitate easy access to the data for analysis and synthesis during the subsequent stages of the review. Key information such as the type of AI models used, the educational outcomes, and implementation challenges will be summarized and categorized for further analysis. Throughout the review process, the research team will continually review and update the data extraction form if new themes or variables emerge that were not initially considered. This flexibility will allow the review to capture a comprehensive range of data, ensuring that the scope of the review remains broad.

## Data synthesis

Data will be mapped and presented in schematic and tabular formats to address the research questions. Alongside these visual representations, a narrative summary of themes will complement the results, describing how the results relate to the review objective and questions. Subsequently, the results will be discussed, highlighting how AI models are used in rehabilitation science education, both successfully and unsuccessfully.

## Discussion

This review will provide a comprehensive synthesis of how AI-driven adaptive learning systems are currently applied in Physical Rehabilitation science education. It will map the range of AI models used in rehabilitation science education. Understanding how these models are integrated into educational practices. The synthesis of these applications will guide curriculum developers in selecting effective AI tools and inform technology designers about the practical needs of the educational domain.

By evaluating reported outcomes this review will highlight the tangible benefits of AI integration. At the same time, it will identify challenges which may hinder equitable access to these technologies. Furthermore, limitations will be examined which will serve as a resource for educators and institutional leaders, equipping them with actionable knowledge to address barriers and optimize the adoption of AI-driven solutions.

The findings from this review will not only deepen the understanding of AI's role in rehabilitation science education but also inspire innovative practices that align technology with the needs of learners and educators. By addressing these critical questions, the review aims

to contribute to the evolution of educational practices, fostering a generation of healthcare professionals equipped to excel in their fields through enhanced learning experiences.

It is important to note that evaluating the quality of the included studies will not be conducted, as this is not required for a scoping review [20,22]. The focus will remain on mapping the breadth of existing literature and identifying key themes, trends, and gaps without assessing the methodological rigor of the individual studies.

## Supporting information

**S1 Appendix. Preferred reporting items for systematic review and meta-analysis extension for scoping review (PRISMA-ScR).**
(ZIP)

**S2 Appendix. PubMed search strategy.**
(DOCX)

## Acknowledgments

We extend our sincere gratitude to the Western University librarians who provided invaluable assistance in refining our search strategy and reviewing the protocol. Their expertise and guidance have been instrumental in ensuring the rigor and comprehensiveness of this work.

## Author contributions

**Conceptualization:** Oyindolapo Olabisi Komolafe, Jannatul Mustofa, Mark Daley, David Walton, Andrews Tawiah.

**Formal analysis:** Oyindolapo Olabisi Komolafe.

**Methodology:** Oyindolapo Olabisi Komolafe, David Walton, Andrews Tawiah.

**Project administration:** David Walton, Andrews Tawiah.

**Supervision:** Mark Daley, David Walton, Andrews Tawiah.

**Writing – original draft:** Oyindolapo Olabisi Komolafe, Jannatul Mustofa.

**Writing – review & editing:** Oyindolapo Olabisi Komolafe, Mark Daley, David Walton, Andrews Tawiah.

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
