## [Decision Letter · Decision Letter 0]

22 Jan 2025

PONE-D-24-57782Current Applications and Outcomes of AI-Driven Adaptive Learning Systems in Rehabilitation Science Education: A scoping review protocolPLOS ONE

Dear Dr. Komolafe,

Thank you for submitting your manuscript to PLOS ONE. After careful consideration, we feel that it has merit but does not fully meet PLOS ONE’s publication criteria as it currently stands. Therefore, we invite you to submit a revised version of the manuscript that addresses the points raised during the review process.

We look forward to receiving your revised manuscript.

Kind regards,

Somayeh Delavari, Ph.D.,

Academic Editor

PLOS ONE

Journal Requirements:

**Additional Editor Comments:**

Dear Authors

Thank you for designing this valuable research project.

comments:

1- Please consider PubMed, Scopus, Pedro and onother specific datases in rehabilitation sciences as additional databases. Medline is subcategory of PubMed; you can delete it.

2- Why is this project mix method?

3- The type of review questions is different in various study design.

https://bjsm.bmj.com/content/55/22/1246.abstract

If there are enough literatures for conducting the present review study, please limit this research project to one of study design category for example observational studies.

4- Please rewrite the background (the abstract) in 3 short sentences: the importance of AI in rehabilitation education, research gap, and the primary outcome

5- If it is possible, please revise review questions according to one of review questions format.

6- Please consider time farmwork and study design as inclusion and exclusion criteria.

7- Please revise key word for searching according to EMTREE and MeSH terms.

Be successful.

Reviewers' comments:

Reviewer's Responses to Questions

**Comments to the Author**

1. Does the manuscript provide a valid rationale for the proposed study, with clearly identified and justified research questions?

Reviewer #1: Yes

Reviewer #2: Yes

Reviewer #3: Yes

2. Is the protocol technically sound and planned in a manner that will lead to a meaningful outcome and allow testing the stated hypotheses?

Reviewer #1: Yes

Reviewer #2: Yes

Reviewer #3: Yes

3. Is the methodology feasible and described in sufficient detail to allow the work to be replicable?

Reviewer #1: Yes

Reviewer #2: Yes

Reviewer #3: Yes

4. Have the authors described where all data underlying the findings will be made available when the study is complete?

Reviewer #1: Yes

Reviewer #2: Yes

Reviewer #3: Yes

5. Is the manuscript presented in an intelligible fashion and written in standard English?

Reviewer #1: Yes

Reviewer #2: Yes

Reviewer #3: Yes

6. Review Comments to the Author

You may also provide optional suggestions and comments to authors that they might find helpful in planning their study.

Reviewer #1: Good luck with your upcoming project, I did not see audiology, rehabilitation nursing and physiatry or rehabilitation medicine in rehabilitation sciences field. Please clarify that. Please check the dicataton of "while" in line 135.

Reviewer #2: Dear authors,

Before all, I would like to thank you for considering this interesting research topic which is now necessary in rehabilitation education. However, I have a few minor comments and questions that require attention and clarification. These are outlined below:

1. Considering that many journals are now indexed in the Scopus database (under Elsevier), I strongly recommend including Scopus as part of the systematic search strategy.

2. Is a quality assessment mandatory in scoping review studies? Please provide clarity on this in the manuscript.

3. In the methodology section, [Line #], kindly introduce JBI as the abbreviation for the Joanna Briggs Instituteupon its first mention.

4. In the discussion section, please elaborate on the rationale for not evaluating the quality of the included studies.

5. Since educational systems vary between countries, it would be helpful to specify the country of each included study and the educational level of participants (e.g., undergraduate or postgraduate students).

Reviewer #3: Thanks for dealing with the education in rehabilitation sciences. Please check the text for spelling (lines 135-136, page 10 of PDF file).

-The education in rehab sciences (in practice) is a broad and diverse field. In case you find enough study on all the disciplines, discussing the results would be challenging.

-Please let the reader know if you include general topics and courses like Anatomy.

-Do you use any AI tool in any part of your study?

7. PLOS authors have the option to publish the peer review history of their article (what does this mean?). If published, this will include your full peer review and any attached files.

Reviewer #1: **Yes: **Holakoo Mohsenifar

Reviewer #2: **Yes: **Mohammad Javaherian

Reviewer #3: No

---

## [Author Response · Author response to Decision Letter 1]

25 Feb 2025

Dear Editorial Team,

Thank you for the editorial comments. The latex template on plos one website is used for formatting the protocol. The following are the responses to the questions raised by the reviewers:

Question: Please consider PubMed, Scopus, Pedro and other specific databases in rehabilitation sciences as additional databases. Medline is subcategory of PubMed; you can delete it & considering that many journals are now indexed in the Scopus database (under Elsevier), I strongly recommend including Scopus as part of the systematic search strategy.

The search database is updated to include Scopus. .The databases searched are Scopus, PubMed, CINAHL, Education Resources Information Center (ERIC), Association for Computing Machinery (ACM), ProQuest Education Journal, Web of Science, ProQuest Dissertations & Theses Global, and IEEE Digital Library

Question: Why is this project mix method?

The scoping review is not a mixed method, thus removed from the abstract

Question: The type of review questions is different in various study design. If there are enough literatures for conducting the present review study, please limit this research project to one of study design category for example observational studies.

This research is a scoping review which typically asks broad questions about the scope of research on a topic. According to Peter et al (2020), a scoping review is used to map concepts, identify and analyse gaps, and inform future research. As such, inclusion of different study designs is encouraged in scoping reviews.

Question: Please rewrite the background (the abstract) in 3 short sentences: the importance of AI in rehabilitation education, research gap, and the primary outcome

The introduction section of the abstract is reduced to meet the requirements

Question: If it is possible, please revise review questions according to one of review questions format.

We have considered the reviewer’s comment, however, because this research is a scoping review, which typically asks broad questions about the scope of research on a topic, we are confident in the diversity of questions. According to Peter et al (2020), a scoping review is used to map concepts, identify and analyse gaps, and inform future research.

Question: Please consider time farmwork and study design as inclusion and exclusion criteria.

The inclusion criteria for a scoping review is contingent on the questions posed. The PCC is stipulated (Population, Concept, and Context). Per Peter (2020) we are attempting to develop a broad understanding of the current state of literature pertaining to our research question. As AI tools in education are a relatively new advancement, we do not anticipate a large volume of research especially when limited to only rehabilitation training contexts. Accordingly, we have an opportunity to conduct a review that is both comprehensive and feasible. We respectfully suggest that retaining the broad search strategy is the best approach for our objectives.

Question: Please revise key word for searching according to EMTREE and MeSH terms.

The search query for Medline and PubMed with the MeSH terms is added to Appendix. The search strategy combines structure database-specific subject headings and keywords/ synonyms of the following concepts:

AI models

Adaptive learning systems

Rehabilitation science (physical therapy, occupational therapy, orthotics, prosthetics, chiropractic, recreation therapy, sports therapy, rehabilitation counseling, kinesiology, cognitive therapy, speech-language pathology and vocational therapy)

Question: Is a quality assessment mandatory in scoping review studies? Please provide clarity on this in the manuscript. In the discussion section, please elaborate on the rationale for not evaluating the quality of the included studies.

In scoping review, critical appraisal (risk of bias assessment) is not required (Munn et al ,2018 Table 1) but reviewers may decide to access and report bias if it is relevant for their objectives. In our case the goal is to determine the current state of literature, what tools are being used and how, rather than assess the quality or risk of bias of the published manuscripts. This will have the added value of enabling our intention of a broad and comprehensive overview while remaining feasible. Paragraph in this regard has been added to discussion with reference to Munn et al. (lines 200-203)

Question: In the methodology section, [Line #], kindly introduce JBI as the abbreviation for the Joanna Briggs Institute upon its first mention.

We have updated the protocol to address the suggestion

Question: Since educational systems vary between countries, it would be helpful to specify the country of each included study and the educational level of participants (e.g., undergraduate or postgraduate students).

Country where study tool place and participants level of education has been added to the Data Extraction table. Thank you for the helpful comment.

Question: The education in rehab sciences (in practice) is a broad and diverse field. In case you find enough study on all the disciplines, discussing the results would be challenging.

The scoping review is focused on physical rehabilitation science which includes: physical therapy, occupational therapy, orthotics, prosthetics, chiropractic, recreation therapy, sports therapy, rehabilitation counseling, kinesiology, cognitive therapy, speech-language pathology and vocational therapy. We are not including psychological / mental health rehabilitation. Our pilot searches indicate the overall volume of relevant literature will not be so large as to make interpretation and discussion problematic.

Question: Please let the reader know if you include general topics and courses like Anatomy.

At this point we are keeping the scope intentionally broad meaning we will include any peer-reviewed research exploring use of any AI tool in any physical rehabilitation training program, regardless of whether that is for a single relevant course like anatomy, or used across a program (e.g., for interactive case studies or learning evaluations). The limiting factor is that the manuscript must explicitly state that the course was included as part of a physical rehabilitation professional training program, rather than a general undergraduate course, for example.

Question: Do you use any AI tool in any part of your study?

No, there is no AI tool used for the scoping review.

Sincerely,

Oyindolapo Komolafe

Western University

References

Munn, Z., Peters, M.D.J., Stern, C. et al. Systematic review or scoping review? Guidance for authors when choosing between a systematic or scoping review approach. BMC Med Res Methodol 18, 143 (2018). https://doi.org/10.1186/s12874-018-0611-x

Peters MDJ, Godfrey C, McInerney P, Munn Z, Tricco AC, Khalil, H. Scoping Reviews (2020). Aromataris E, Lockwood C, Porritt K, Pilla B, Jordan Z, editors. JBI Manual for Evidence Synthesis. JBI; 2024. Available from: https://synthesismanual.jbi.global. https://doi.org/10.46658/JBIMES-24-09

---

## [Decision Letter · Decision Letter 1]

24 Mar 2025

PONE-D-24-57782R1Current Applications and Outcomes of AI-Driven Adaptive Learning Systems in Physical Rehabilitation Science Education: A scoping review protocolPLOS ONE

Dear Dr. Komolafe,

Thank you for submitting your manuscript to PLOS ONE. After careful consideration, we feel that it has merit but does not fully meet PLOS ONE’s publication criteria as it currently stands. Therefore, we invite you to submit a revised version of the manuscript that addresses the points raised during the review process.

We look forward to receiving your revised manuscript.

Kind regards,

Somayeh Delavari, Ph.D.,

Academic Editor

PLOS ONE

Journal Requirements:

Additional Editor Comments:

Dear Athors

Thank you so much for revising the manuscript. Please check the line 136; you repeated "AND" for twice.

Kind Regards.

Reviewers' comments:

Reviewer's Responses to Questions

**Comments to the Author**

Review Comments to the Author

You may also provide optional suggestions and comments to authors that they might find helpful in planning their study.

Reviewer #1: After thoroughly reviewing this manuscript, all of the questions have been addressed, and it has been accepted.

Reviewer #3: -Please replace "Rehabilitation Sciences" instead of "Rehabilitation Science" in the title and manuscript.

-Please let the readers know why your search didn't include optometry and audiology.

-You have focused on "Physical rehabilitation". Please let the readers know why you include cognitive therapy.

-Although "adaptive learning" has been considered in the process, the main questions do not reflect it.

-Please define education clearly; does it also include evaluation and assessment (It should be).

---

## [Author Response · Author response to Decision Letter 2]

30 Apr 2025

Question: Please let the readers know why your search didn't include optometry and audiology.

Audiology has been included in the search strategy. The definition of Physical Rehabilitation is stated to justify the inclusion of the disciplines (lines 22-28)

Question: You have focused on "Physical rehabilitation". Please let the readers know why you include cognitive therapy.

Cognitive therapy has been removed from the protocol and search strategy

Question: Although "adaptive learning" has been considered in the process, the main questions do not reflect it

As outlined in the introduction (lines 1-8), we define adaptive learning as the use of AI-driven technology to personalize the learning experience, including aspects such as content delivery, evaluation, and assessment. Our intention with the current formulation of the research questions is to take a broad and inclusive approach to understanding how AI technologies—particularly those with adaptive capabilities—are being utilized within physical rehabilitation science education. -While the term adaptive learning is not explicitly mentioned in the research questions, it is conceptually embedded in our first question, which explores the types of AI models and their applications in enhancing teaching and learning. This phrasing was chosen to capture both general AI applications and those that include adaptive learning features such as personalization, real-time feedback, and learner-specific modifications.

Question: Please define education clearly; does it also include evaluation and assessment (It should be).

Thank you for this thoughtful observation. We agree that education, particularly in the context of adaptive learning, should be clearly defined to encompass not only instructional delivery but also evaluation and assessment. To address this, we have revised the introduction to explicitly define education as a multidimensional process that includes teaching, learning, assessment, and feedback. This definition aligns with the mechanisms of adaptive learning systems, which rely heavily on continuous assessment to personalize learning experiences.

---

## [Decision Letter · Decision Letter 2]

18 May 2025

Current Applications and Outcomes of AI-Driven Adaptive Learning Systems in Physical Rehabilitation Science Education: A scoping review protocol

PONE-D-24-57782R2

Dear Dr. Komolafe,

We’re pleased to inform you that your manuscript has been judged scientifically suitable for publication and will be formally accepted for publication once it meets all outstanding technical requirements.

Kind regards,

Somayeh Delavari, Ph.D.,

Academic Editor

PLOS ONE

---

## [Editor Report · Acceptance letter]

PONE-D-24-57782R2

PLOS ONE

Dear Dr. Komolafe,

I'm pleased to inform you that your manuscript has been deemed suitable for publication in PLOS ONE. Congratulations! Your manuscript is now being handed over to our production team.

Kind regards,

on behalf of

Dr. Somayeh Delavari

Academic Editor

PLOS ONE